# Anemia status and its determinants among reproductive-age women in Tanzania: A multi-level analysis of Tanzanian demographic and health survey data

Gizachew Ambaw Kassie[1]*, Aklilu Habte Hailegebireal[2,3], Amanuel Yosef Gebrekidan[4], Beshada Zerfu Woldegeorgis[5], Getachew Asmare Adella[6], Kirubel Eshetu Haile[7], Yordanos Sisay Asgedom[1]

1 Department of Epidemiology and Biostatistics, College of Health Sciences and Medicine, Wolaita Sodo University, Wolaita Sodo, Ethiopia, 2 Faculty of Health and Environmental Sciences, Auckland University of Technology, Auckland, New Zealand, 3 School of Public Health, College of Medicine and Health Sciences, Wachemo University, Hosanna, Ethiopia, 4 School of Public Health, College of Health Science and Medicine, Wolaita Sodo University, Wolaita Sodo, Ethiopia, 5 School of Medicine, College of Health Science and Medicine, Wolaita Sodo University, Wolaita Sodo, Ethiopia, 6 Department of Reproductive Health and Nutrition, College of Health Science and Medicine, Wolaita Sodo University, Wolaita Sodo, Ethiopia, 7 School of Nursing, College of Health Science and Medicine, Wolaita Sodo University, Wolaita Sodo, Ethiopia

* gizachewambawkase@gmail.com

**Data Availability Statement:** All relevant data are within the manuscript.

## Abstract

### Introduction

Anemia is significantly associated with morbidity and mortality in women of reproductive age. Determining the prevalence and identifying associated risk factors remains an important topic in public health. Therefore, this study aimed to estimate and identify the risk factors for anemia in women of reproductive age in Tanzania.

### Materials and methods

A cross-sectional study utilizing secondary data from the Tanzania Demographic and Health Survey of 2022 was conducted. Weighted total samples of 8,921 reproductive-age women were included in the analysis. A multilevel Poisson regression model was employed to account for the hierarchical structure of the TDHS data. Deviance was used to compare the models. In the multivariable multilevel mixed-effects Poisson regression model with robust variance, adjusted prevalence ratios with corresponding 95% confidence intervals were used to identify the determinants of anemia.

### Results

The prevalence of anemia among women of reproductive age in Tanzania was 42.02% [95% CI: (40.79%–43.25%)]. Of these, 19.82%, 19.35%, and 2.84% had mild, moderate, and severe anemia, respectively. The regression results revealed that women from households with the poorest wealth quantiles, underweight, pregnant status, high community poverty level, and women from the eastern and Zanzibar administrative zones had a higher

**Funding:** The author(s) received no specific funding for this work.

**Competing interests:** The authors have declared that no competing interests exist.

**Abbreviations:** APR, Adjusted Prevalence Ratio; CI, Confidence Interval; EAs, Enumeration areas; ICC, Intra-cluster Correlation Coefficient; IR, Individual Record; LLR, Log likelihood ratio; LR, Likelihood ratio; SDG, Sustainable Development Goal; SSA, sub-Saharan Africa; TDHS, Tanzanian Demographic health survey; WHO, World Health Organization.

prevalence of anemia. While currently employed, moderate alcohol consumption and use of hormonal contraceptive methods were associated with a lower prevalence of anemia.

## Conclusion

The findings of the study showed that anemia is a severe public health issue among women of reproductive age in Tanzania, affecting more than four in ten women. Interventions targeting the improvement of nutrition, access to healthcare services, and education on anemia prevention and management should be prioritized to reduce the burden of anemia effectively.

## Introduction

Anemia affects millions of women of reproductive age worldwide and is characterized by low levels of healthy red blood cells and hemoglobin, which reduces the ability of the blood to transport oxygen to body tissues [1,2]. Because of the physiological state of menstruation, pregnancy, and lactation, it is the most common nutritional problem disproportionately affecting women of reproductive age [3]. During each stage of a woman's reproductive cycle, there is a varying risk of exposure to anemia, but its presence throughout all stages is associated with poor outcomes for the mother, newborn, and children [4–6]. Pregnant women who suffer from anemia during the preconception period are at an increased risk of experiencing spontaneous abortions, giving birth to infants with low birth weights, and restricting fetal growth. Furthermore, these children may be more likely to develop long-term disorders, such as autism and intellectual disabilities [7,8].

The global incidence of anemia is 30% among non-pregnant women and 36% among pregnant women [9]. Approximately 32% of childbearing women in low- and middle-income countries (LMICs) suffer from anemia. More than 50% of these countries have anemia levels between 20% and 39.9% in this age group [10]. It is most prominent in SSA countries with a prevalence of 40.5%. In Africa, the prevalence of anemia varies widely from 13% in Rwanda to 62% in Mali [11]. According to data from 2015-16Tanzania Demographic and Health Survey, anemia is alarmingly prevalent among Tanzanian women of reproductive age (45%) [2]. Iron deficiency is a serious dietary problem that accounts for 50% of instances of anemia worldwide. It is the primary contributor to anemia. Nevertheless, deficiencies in vitamin A, vitamin B12, and folic acid can also result in nutritional deficiency anemia. Acute and chronic inflammations, parasite infections, and hereditary or acquired diseases affecting hemoglobin synthesis and the creation or survival of red blood cells can also be causes of anemia [3].

Anemia in during pregnancy is linked to several unfavorable health consequences, such as an elevated risk of maternal and infant mortality, preterm delivery, low birth weight, and impaired cognitive development in offspring. These devastating impacts make anemia in pregnant mothers to be one of the global health priority areas at the global level, especially in resource-limited areas [12,13].

Evidences revealed that anemia is attributed to a complicated interplay of factors, including dietary deficiencies, parasitic infections, and socioeconomic status. Particularly iron deficiency, infectious diseases like malaria and hookworm infestation, limited access to healthcare and prenatal care, cultural practices such as early marriage and teenage pregnancy, as well as socioeconomic factors like poverty and lack of education [2,14].

To reduce anemia in women of reproductive age several interventions have been implemented such as, increasing access to iron supplements, promoting iron-rich diets, strengthening antenatal care, and educating the public about reproductive health. Despite these efforts, anemia rates among non-pregnant and pregnant women have declined by only 1% and 5% over the last two decades, respectively [9]. Furthermore, existing evidence shows that none of the LMICs, including Tanzania, are on track to achieve the nutrition target of 50% anemia reduction by 2030, indicating that the average annual rate of decline was below the level needed [15]. Thus, for effective intervention strategies and policy development in Tanzania, it is essential to generate up-to-date national, sub-regional, and regional estimates of anemia and to identify individual- and community-level factors[16]. As a result of, understanding the determinants of anemia among reproductive-age women in Tanzania is crucial for developing targeted interventions and policies to address this public health challenge.

To date, limited evidence exists in Tanzania regarding anemia and its determinants, with most studies conducted in a single setting lacking multilevel modeling to consider the hierarchical structure of the data from the Demographic and Health Survey (DHS). Moreover, the data were not weighted to compensate for the non-proportional distribution of the sample size and proper standard error estimation. Additionally, using a multilevel analysis approach can help to account for the complex interplay of individual-, household-, and community-level factors that may contribute to anemia prevalence, providing a more comprehensive understanding of the issue. Therefore, this study aimed to conduct a multi-level analysis of data from the Tanzanian Demographic and Health Survey (TDHS) to identify the individual, household, and community-level factors associated with anemia status among reproductive-age women.

The multi-level analysis approach is particularly relevant in this context, as it allows for the examination of the relative contributions of individual, household, and community-level factors to the prevalence of anemia. In addition, this research will provide updated and reliable data on the determinants of anemia among women of reproductive age in Tanzania in designing targeted interventions and programs to reduce the burden of anemia.

## Method and materials

### Study design, data source, population and sampling procedure

A nationally representative, cross-sectional study utilizing secondary data from the Tanzania Demographic and Health Surveys (TDHS) of 2022 was conducted. The TDHS is conducted every five years to collect updated health and health-related information. The 2022 TDHS sample design was carried out in two stages, with the purpose of providing national, urban, and rural estimates for Tanzania Mainland and Zanzibar. For specific indicators, such as contraceptive use, the sample design allows for estimates at the regional level for each of the 31 regions—26 regions in Tanzania Mainland and s5 regions in Zanzibar.

The 2022 TDHS employed a stratified, two-stage sample design. The first stage involved selecting sampling points (clusters) comprised of enumeration areas (EAs) delineated for the 2012 Tanzania Population and Housing Census (2012 PHC). Enumeration areas were selected using probability proportional to size (PPS) sampling method. The grouping of regions into zones allowed for larger denominators and smaller sampling errors for indicators at the zonal level. Tanzania was divided into several strata based on geographical regions and urban/rural areas. Each stratum was treated as a separate sampling domain. A total of 629 clusters were selected, 211 urban and 418 rural. In the second stage, 26 households were systematically chosen from each cluster, resulting in a total sample size of around 16,354 households. To account for the unequal probability of selection and non-response, individual weights were calculated for each respondent. For this study, we utilized the individual record (IR) file sourced from

datasets within the TDHS 2022, which comprised information from 15,254 women aged 15–49 years. Hemoglobin test was conducted for the subsample of 7577 of reproductive age women. Finally, a total weighted sample size of 8,921 reproductive-age women (unweighted sample = 7,577) was considered for this analysis (Fig 1). The specifics of the sampling technique, sample size, data collection instruments, data quality control measures, and ethical considerations are all outlined in the 2022 TDHS report [17].

## Study variables and measurements

**Dependent variable.** This research was conducted using altitude-adjusted hemoglobin (Hgb) levels from the DHS data, which provided the foundation for the study. The outcome variable measured was the anemia level. In TDHS, anemia level was assessed using Hgb level adjusted for altitude and pregnancy status. In this regard, a Hgb level of 12 g/dL, and 11 g/dL was considered as the cut-off point to determine anemia among non-pregnant and pregnant women, respectively. For non-pregnant women, mild anemia is classified as having hemoglobin (Hgb) levels between 11.0 and 11.9 g/dL, moderate anemia is between 8.0 and 10.9 g/dL, and severe anemia is when Hgb levels are less than 8.0 g/dL. For pregnant women, mild anemia is classified as having Hgb level between 10.0 and 10.9 g/dL, moderate anemia is between 7.0 and 9.9 g/dL, and severe anemia is when Hgb level is less than 7.0 g/dL. [18]. For this study, the anemia levels were re-categorized from the previous classifications (no, mild, moderate, and severe) as anemic being coded as "1" and non-anemic being coded as "0."

**Independent variables.** Based on a literature review, we considered both individual-level and community-level variables. To account for the hierarchical nature of the DHS data, two sources of independent variables were used in the study (individual and community-level variables). Age of respondents, maternal education, marital status, occupation of respondent, sex of household head, number of household members, household wealth status, health insurance, parity, body mass index, modern contraceptive use, current pregnancy status, history of pregnancy termination, breastfeeding, internet use and alcohol consumption were level one variable. Residence, administrative zone, community poverty, and community maternal education were level-two variables. These two community-level variables (community maternal education and poverty) were generated by aggregating maternal education and household wealth status at the cluster/enumeration area levels. They were then categorized as having higher community maternal education and poverty based on the national median value of maternal education and poverty since they were not normally distributed. A community's poverty level was categorized as high if 75–100% of its women were in the two lowest wealth quintiles, as moderate if the proportion was 50–74%, and as low if it was 0–49%. The level of community illiteracy was determined by using the percentage of illiterate women per cluster; if it was under 49%, the level was classified as low, if it was between 50 and 74%, it was classified as moderate, and if it was 75–100%, it was classified as high. Both individual and community level variables was derived from a comprehensive search of various literatures. Body mass index was classified as underweight (BMI < 18.5 kg/m2), normal weight (18.50–24.99 kg/m2), overweight (25.0–29.9 kg/m2), and obese ($\geq$ 30.0 kg/m2) according to World Health Organization (WHO) cutoff criteria [19].

## Data management and analysis

A preliminary check of the data was performed to determine if there were any missing values, outliers, or inconsistencies. A descriptive and analytical analysis of the data was carried out using STATA software, which was used for data extraction, further recoding, and further analysis. Before conducting any statistical analysis, the data underwent weighting using the

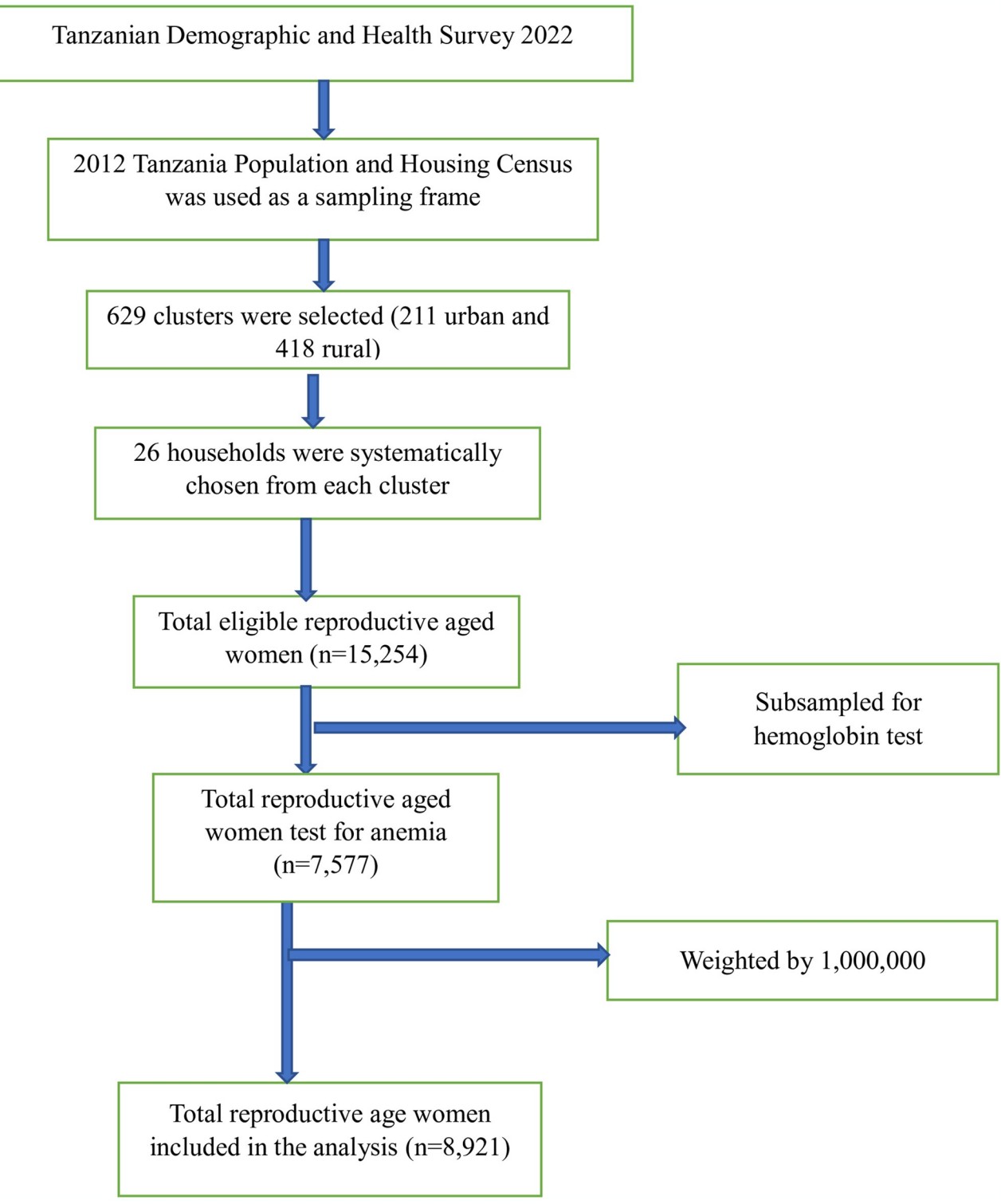

**Fig 1. Sampling procedure and sampling technique anemia status and its determinants among reproductive-age women in Tanzania using 2022 TDHS.**

sampling weight, primary sampling unit, and stratum to ensure the survey's representativeness and account for the sampling design when computing standard errors. This was done to produce accurate statistical estimates. Frequencies and percentages were used for descriptive analysis. Considering the hierarchical nature of the DHS data, a multilevel model was fitted.

A multilevel Poisson regression model with robust variance was employed to identify factors associated with anemia, and the results were presented as a prevalence ratio with a 95% CI. We chose this model because the odds ratio estimated using logistic regression in a cross-sectional study can significantly overestimate the relative risk when the outcome is common [20]. To check whether there was a clustering effect or variability, the intra-class correlation coefficient (ICC), the proportional change in variance (PCV), and the median odds ratio (MOR) were reported.

In the bi-variable analysis, variables with p-values > 0.20 were selected for the multivariable analysis. Four models were fitted during the multilevel analysis: the null model (containing only outcome variables), model I (containing only individual level variables), model II (containing only community level variables), and model III (incorporating both individual and community level variables simultaneously). Deviances were used in model comparisons, and a prevalence ratio with a 95% confidence interval (CI) was reported for the best-fitting model. Lastly, variables with p-values of 0.05 in the multivariable multilevel Poisson regression analysis were considered significant factors associated with anemia prevalence in women.

## Data collection procedure

Blood samples were collected from eligible women who consented to the test and were tested onsite for anemia using a battery-operated portable HemoCue analyzer, with results provided verbally and in writing. The recoded files were adjusted for altitude, with the hemoglobin count being lowered as altitude increased, based on the following formula: Adjust adjHb if adjusted > 0, where adjust is the adjustment amount, alt is altitude in 1,000 feet (converted from meters), and adjHb is the adjusted hemoglobin level. No adjustment was made for altitudes below 1,000 meters, and both the adjusted and unadjusted hemoglobin counts are included in the recode files.

## Ethical consideration

The present investigation was not obliged to seek participant consent or ethical clearance, as it involved a secondary analysis of publicly accessible survey data obtained from the MEASURE DHS program. The relevant website (<http://www.dhsprogram.com>) granted us permission to download and employ the data for our research purposes. The dataset did not comprise residential addresses or personal names.

## Results

### Study participant's descriptive characteristics

An overall weighted sample of 8,921 women of reproductive age was used for this study. Among the study participants, the median age was 28 (IQR = 21–37) years, and the majority (38.5%) were between 15 and 24 years old. A quarter of participants (2,368 (26.5%)) came from households with the highest wealth quintile, and 5,583 (62.6%) were employed. In terms of the gender of household heads, 6,276 (70.4%) of respondents were male. The majority of our study participants, 4,370 (48.9%), had a primary education, and 4,005 (44.9%) were married. There were 3,374 (37.8%) multiparous respondents and 2,527 (28.3%) nulliparous respondents. Among the respondents, 2,035 (22.8%) were lactating, and 669 (7.5%) were

pregnant. Twenty-eight percent (28.9%) of the 2,581 respondents currently use hormonal or non-hormonal contraception. A majority of 8,413 (94.3%) respondents were from households with health insurance, and only 2,581 (28.9%) use internet (Table 1).

## Community-level characteristics of the study participants

Of the study participants, 5,710 (64%) were from rural areas, 2,016 (22.5%) were from the eastern administrative zone, 558 (6.2%) were from the south, and 1,334 (15.0%) were from Zanzibar. Community maternal education was high in two-thirds of 2,918 (32.7%) of the study participants, and the level of poverty was low in 5,127 (57.5%) of the community (Table 2).

## Prevalence of anaemia among reproductive age women in Tanzania

The overall prevalence of anemia among Tanzanian reproductive-age women was 42.02% (95% CI: 40.79%, 43.25%). Of these 19.82 (95% CI: 18.38%, 20.36%) of reproductive age women had mild anemia, moderate anemia 19.35% (95% CI: 18.86%, 20.82%), and 2.84% (95% CI: 2.46%, 3.28%) had severe anemia.

## Random effect and model fitness analysis results

In this study, a single-level binary Poisson regression model was compared to a multilevel robust Poisson regression model without predictors. The deviance (-2 Likelihood Ratio (LR) test) was used to determine which model was more appropriate, and the result was statistically significant ($p<0.05$), suggesting that the multilevel robust Poisson regression model with robust variance was a better fit than the single-level robust Poisson regression model. Four random effects models were fitted, and the model with the lowest deviation was selected. Furthermore, an intra-class correlation coefficient (ICC) of 0.1034 was observed in the null model, suggesting that 10.34% of anemia was influenced by unobserved factors at community levels. The Akaike's Information Criteria (AIC) is the smallest in model 4 (AIC = 13742.0) as compared to other models. Thus, model 4 is the best-fitting model. Therefore, all interpretations and reports were made based on this model (Table 3).

## Determinants of anaemia among reproductive age women

In the multivariable multilevel mixed-effect Poisson regression model, poorest household wealth index, being employed, being pregnant, hormonal contraceptive use, underweight, alcohol consumption, and women from (eastern, central, southern highland and Zanzibar) administrative zones and high community poverty level were predictors of anemia among reproductive-age women. It was found that anemia prevalence was 1.07 higher among unemployed reproductive-age women compared with employed women [APR = 1.07, 95% CI: 1.01, 1.16]. Reproductive-age women from the poorest household wealth index had 1.17 times [AOR = 1.17, 95% CI: 1.12, 1.57] higher prevalence of anemia as compared to reproductive-age women from the richest household wealth index. Women with a body mass index of <18.5 kg/m2 (underweight) had 1.15 times higher prevalence of anemia than women who had BMI ≥18.5 kg/m2 [APR = 1.15, 95% CI: 1.09, 1.82]. Women who are using hormonal contraceptives were 28% less likely to be anemic compared to those who do not using [APR = 0.72, 95% CI: 0.64, 0.81]. Being a pregnant had 1.21 times higher prevalence of anemia as compared to non-pregnant women [APR = 1.21, 95% CI: 1.1, 1.34]. Women of reproductive age who did not drink alcohol had 1.29 times higher prevalence of anemia than women who did [AOR = 1.29; 95% CI: 1.08, 1.52].

**Table 1. Individual-level characteristics of reproductive age women Tanzanian demographic and health survey, 2022.**

| Variables | Category | Weighted frequency (Anaemia status) | | Total weighted frequency (%) |
|---|---|---|---|---|
| | | Not anaemic (%) | Anaemic (%) | |
| **Maternal age in years** | 15–24 | 1,896 (54.9) | 1,554 (45.1) | 3,450 (38.7%) |
| | 25–34 | 1,590 (60.2) | 1,051 (39.8) | 2,641 (29.6%) |
| | ≥35 | 1,687 (59.6) | 1,143 (40.4) | 2,829 (31.7%) |
| **Household wealth status** | Poorest | 714 (55.9) | 563 (44.1) | 1,276 (14.3%) |
| | Poorer | 930 (62.5) | 557 (37.5) | 1,487 (16.7%) |
| | Middle | 1,110 (60.9) | 711 (39.1) | 1,822 (20.4%) |
| | Richer | 1,087 (55.3) | 880 (44.7) | 1,968 (22.1%) |
| | Richest | 1,332 (56.3) | 1,036 (43.7) | 2,368 (26.5%) |
| **Sex of household head** | Male | 3,660 (58.3) | 2,616 (41.7) | 6,276 (70.4%) |
| | Female | 1,512 (57.2) | 1,133 (42.8) | 2,645 (29.6%) |
| **Women's occupation** | Unemployed | 1,806 (54.1) | 1,531 (45.9) | 3,337 (37.4%) |
| | Employed | 3,366 (60.3) | 2,217 (39.7) | 5,583 (62.6%) |
| **Maternal educational status** | No education | 796 (56.3) | 617 (43.7) | 1,413 (15.8%) |
| | Primary | 2,654 (60.7) | 1,716 (39.3) | 4,370 (48.9%) |
| | Secondary | 1,661 (54.9) | 1,364 (45.1) | 3,025 (32.7%) |
| | Higher | 62 (54.9) | 51 (45.1) | 113 (1.5%) |
| **Current Marital status** | Never married | 1,376 (53.9) | 1,175 (46.1) | 2,551 (28.6%) |
| | Married | 2,341 (58.4) | 1,665 (41.6) | 4,005 (44.9%) |
| | widowed/divorced | 1,456 (61.6) | 908 (38.4) | 2,364 (26.5%) |
| **Body mass index** | Normal | 2,917 (56.4) | 2,256 (43.6) | 5,174 (58.0%) |
| | Underweight | 431 (52.9) | 383 (47.1) | 814 (9.1%) |
| | Overweight | 1,058 (59.6) | 717 (40.4) | 1,774 (19.9%) |
| | Obese | 766 (66.1) | 393 (33.9) | 1,159 (16%) |
| **Parity** | Nulliparous | 1,305 (51.6) | 1,222 (48.4) | 2,527 (28.3%) |
| | Primi-parous | 794 (57.4) | 589 (42.6) | 1,383 (15.4%) |
| | Multi-parous | 2,090 (61.9) | 1,284 (38.1) | 3,374 (37.8%) |
| | Grandmulti-parous | 983 (60.1) | 653 (39.9) | 1,636 (18.3%) |
| **Currently pregnant** | Yes | 308 (46.1) | 360 (53.9) | 669 (7.5%) |
| | No or unsure | 4,864 (58.9) | 3,389 (41.1) | 8,253 (92.5%) |
| **Utilization of contraceptives** | No using | 3,463 (54.6) | 2,877 (45.4) | 6,340 (71.1%) |
| | Pills/implant/injectable | 1,221 (71.7) | 481 (28.3) | 1,702 (19.1%) |
| | non hormonal | 488 (55.5) | 391 (44.5) | 879 (9.8%) |
| **Currently breast feeding** | Yes | 1,266 (62.2) | 769 (37.8) | 2,035 (22.8%) |
| | No | 3,906 (56.7) | 2,979 (43.3) | 6,885 (77.2%) |
| **Currently amenorrheic** | Yes | 678 (59.6) | 459 (40.4) | 1,137 (12.7%) |
| | No | 4,494 (57.7) | 3,289 (42.3) | 7,784 (87.3%) |
| **History of pregnancy termination** | Yes | 676 (53.8) | 579 (46.2) | 1,255 (14.1%) |
| | No | 4,496 (58.7) | 3,169 (41.3) | 7,665 (85.9%) |
| **Alcohol consumption** | Yes | 387 (71.1) | 158 (28.9) | 545 (6.1%) |
| | No | 4,786 (57.1) | 3,590 (42.9) | 8,376 (93.9%) |
| **Internet use** | Never | 3,463 (54.6) | 2,877 (45.4) | 6,340 (71.1%) |
| | Yes | 1,709 (66.2) | 872 (33.8) | 2,581 (28.9%) |
| **Health insurance** | Yes | 4,867 (57.9) | 3,546 (42.1) | 8,413 (94.3%) |
| | No | 305 (60.0) | 203 (40.0) | 508 (5.7%) |

**Table 2. Community-level characteristics of reproductive age women Tanzanian demographic and health survey, 2022.**

| Variables | Category | Weighted frequency (Anaemia status) | | Total weighted frequency (%) |
|---|---|---|---|---|
| | | Not anaemic (%) | Anaemic (%) | |
| **Place of Residence** | Urban | 1,849 (57.6) | 1,361 (42.4) | 3,211 (36%) |
| | Rural | 3,323 (58.2) | 2,387 (41.8) | 5,710 (64%) |
| **Administrative zones of Tanzania** | Central | 597 (66.7) | 298 (33.3) | 896 (10.1%) |
| | Southern | 369 (66.1) | 189 (33.9) | 558 (6.2%) |
| | Southwest highland | 564 (69.1) | 252 (30.8) | 816 (9.1%) |
| | Eastern | 1,052 (52.2) | 964 (47.8) | 2,016 (22.5%) |
| | Southern highland | 599 (71.0) | 245 (29.0) | 844 (9.5%) |
| | Western | 567 (60.2) | 375 (39.8) | 942 (10.6%) |
| | Lake | 857 (56.6) | 658 (43.4) | 1,515 (17.0%) |
| | Zanzibar | 565 (42.4) | 769 (57.6) | 1,334 (15.0%) |
| **Community maternal education** | Low | 1,805 (58.3) | 1,290 (41.7) | 3,096 (34.7%) |
| | Moderate | 1,696 (58.3) | 1,211 (41.65) | 2,907 (32.6%) |
| | High | 1,672 (57.3) | 1,246 (42.7) | 2,918 (32.7%) |
| **Community poverty** | Low | 2,914 (56.8) | 2,213 (43.2) | 5,127 (57.5%) |
| | Moderate | 520 (59.9) | 347 (40.1) | 867 (9.7%) |
| | High | 1,739 (59.4) | 1,188 (40.6) | 2,927 (32.8%) |

**Table 3. Model summary of multilevel robust Poisson regression analysis of factors associated with anemia among reproductive age women Tanzanian demographic and health survey, 2022.**

| Parameter | Null model | Model I | Model II | Model III |
|---|---|---|---|---|
| Intra-class correlation coefficient | 0.1034 | 0.0872 | 0.0544 | 0.0529 |
| Variance (Standard Error) | 0.029 (0.021) | 0.018 (0.014) | 0.026 (0.012) | 0.019 (0.011) |
| Deviance | 11985.9 | 13790.94 | 13794.2 | 11672.1 |
| Proportional change in variance | Reference | 0.3148 | -0.3684 | 0.3448 |
| Akaike Information Criteria | 13990.92 | 13836.94 | 13822.2 | 13742.09 |
| Bayesian Information Criteria | 12004.78 | 13996.4 | 13919.26 | 13984.74 |
| -Log Likelihood Ratio | 5993.45 | 6895.474 | 6897.19 | 5836.04 |
| LR-test | LR test vs. logit model: chibar2(01) = 202.03 Prob > = chibar2 <0.001 | | | |

The prevalence of anemia was 1.16 times higher among reproductive age women from eastern administrative zones [APR = 1.16, 95% CI: 1.05, 1.28] and 1.36times higher women from Zanzibar administrative zones of Tanzania [APR = 1.36, 95% CI: 1.21, 1.51]. Whereas being anemic was 27% less likely among women from southern highland administrative zones [APR = 0.73, 95% CI: 0.62, 0.86] and 12% less likely among women from central administrative zone [APR = 0.78, 95% CI: 0.66, 0.93] as compared to reproductive age women from lake administrative zone and in southern [AOR = 0.73, 95% CI: 0.62, 0.82]. Being anemic was 1.13 times more likely among reproductive-age women from high community poverty levels as compared to reproductive-age women from low community poverty levels [APR = 1.13, 95% CI: 1.01, 1.25] (Table 4).

## Discussion

Anemia is a global health problem, particularly prevalent among women of reproductive age in low-income countries. Tanzania, like many other low-income countries, grapples with a

**Table 4. Multi-variable multilevel mixed-effects robust Poisson regression analysis of factors associated with anemia among reproductive age women Tanzanian demographic and health survey, 2022.**

| Variables | Category | Model I (level I variables) APR (95% CI) | Model II (level II variables) APR (95% CI) | Model III (level 1 and level 2 variables), APR (95% CI) |
|---|---|---|---|---|
| Women's age in years | 15–24 years | 0.96 (0.879, 1.07) | | 0.98 (0.88, 1.08) |
| | 25–34 years | 0.97 (0.89, 1.1.06 | | 0.97 (0.88, 1.06) |
| | > = 35 years | 1 | | 1 |
| Educational status | No education | 1 | | 1 |
| | Primary | .91 (0.83, 1.00) | | 0.95 (0.87, 1.04) |
| | Secondary | 0.93 (0.84, 1.03) | | 0.90 (0.81, 1.00) |
| | Higher | 0.93 (0.70, 1.21 | | 0.94 (0.71, 1.23) |
| Marital status | Never married | 1 | | 1 |
| | Married | 1.01 (0.92, 1.12) | | 0.96 (0.87, 1.06) |
| | Separated/divorced | 0.99 0.89, 1.1 | | 0.99 (0.94, 1.89) |
| Occupation | Employed | 1 | | |
| | Unemployed | 1.08 (1.01, 1.15) | | 1.07 (1.01, 1.16) |
| Parity | Null-parus | 1 | | 1 |
| | Multip-parus | 0.97 (0.87, 1.08) | | 0.82 (0.82, 1.04) |
| | Grand Multiparus | 1.12 (0.94, 1.76) | | 0.91 (0.78, 1.04) |
| Household wealth status | Poorest | 1.18 (1.13, 1.62) | | **1.17 (1.12, 1.57)*** |
| | Poor | 1.07 (0.98, 1.16) | | 1.08 (0.99, 1.18) |
| | Middle | 0.93 (0.85, 1.03) | | 097 (0.86, 1.08) |
| | Rich | 0.91 (0.81,1.02) | | 0.96 (0.83, 1.11) |
| | Richest | 1 | | 1 |
| BMI | <18.5 kg/m$^2$ | 1.18 (1.09, 1.86) | | **1.15 (1.09, 1.82)*** |
| | < = 18.5 kg/m$^2$ | 1 | | 1 |
| Contraceptive use | Not using | 1 | | 1 |
| | Non hormonal | 1.07 (0.97, 1.18) | | 1.08 (0.98, 1.19) |
| | Hormonal | 0.69 (0.62, 0.77) | | **0.72 (0.64, 0.81)*** |
| Pregnancy | Yes | 1.22 (1.1, 1.34) | | **1.21 (1.1, 1.34)*** |
| | No | 1 | | 1 |
| Breast feeding | Yes | 0.94 (0.86, 1.02) | | 0.94 (0.87, 1.02) |
| | No | 1 | | 1 |
| Alcohol use | Yes | 1 | | 1 |
| | No | 1.33 (1.12, 1.58) | | **1.29 (1.08, 1.52)*** |
| Place of Residence | Rural | | 0.98 (0.89, 1.08) | 1.01 (0.92, 1.11) |
| | Urban | | 1 | 1 |
| Administrative zones | Lake | | 1 | 1 |
| | Eastern | | 1.14 (1.03, 1.26) | **1.16 (1.05, 1.28)*** |
| | Central | | 0.77 (0.65, 0.91) | **0.78 (0.66, 0.93)*** |
| | Southern | | 0.78 (0.63, 0.96) | 0.84 (0.68, 1.03) |
| | Western | | 0.88 (0.75, 1.03) | 0.88 (0.75, 1.03) |
| | Southwestern highland | | 0.71 (0.58, 0.88) | 0.73 (0.59, 0.92) |
| | Southern highland | | 0.69 (0.59, 0.81) | **0.73 (0.62, 0.86)*** |
| | Zanzibar | | 1.4 (1.26, 1.56) | **1.36 (1.21, 1.51)*** |
| Community poverty | Low | | 1 | 1 |
| | Moderate | | 0.91 (0.79, 1.04) | 1.02 (0.93, 1.11) |
| | High | | 1.1 (1.03, 1.27) | **1.13 (1.01, 1.25)*** |
| Community education | Low | | 1.02 (0.91, 1.14) | 0.98 (0.87, 1.12) |
| | Moderate | | 1.03 (0.94, 1.12) | 0.92 (0.11, 1.05) |
| | High | | 1 | 1 |

high prevalence of anemia among its population, posing adverse consequences for individuals and society at large. This research aimed to shed light on the prevalence, contributing factors, and implications of anemia in Tanzania.

According to this study, Tanzania faces a significant burden of anemia, with a prevalence rate of 42.02% among reproductive-age women. The finding from this study was consistent with the studies from Sub-Saharan Africa (40.5%) [11], low-middle income countries (41.58%) [21] and Gambia (44.28%) [22]. However, the finding from this study is higher than the prevalence reported in Ethiopia (37.4%) [23], loa peoples of democratic republic (39.2%) [24], Philippines (9.0%) [25], Rwanda (19.2%) [13], Mali (38%) [26], east Africa (34.85%) [27] and systematic review in SSA (32.1%) [28]. On the other hand the finding from the current study was lower than the study conducted in Nepal 64.32% [29], Senegal 55.2% [12], Mali (63.5%) [30], Pakistan 61.3% [31] and India (65.5%) [32]. The potential reason may be due to differences in socioeconomic status among countries such as chronic malnutrition, limited access to nutritious food, poor sanitation and hygiene, inadequate healthcare infrastructure and lack of education and awareness [33]. Addressing these issues through strategies such as improving food access, promoting hygiene practices, and enhancing healthcare infrastructure can reduce anemia in Tanzania.

In this study, determinants of anemia were also identified. In the final model, we found that Poorest Wealth index, Being employed, BMI $<18.5$ kg/m$^2$, hormonal contraceptive use, pregnancy status, alcohol consumption, administrative zones and high community poverty level were significantly associated with anemia among reproductive age women.

Women from families with poorest household wealth index had increased the prevalence ratio of anemia by 7% as compared to women from richest households. This is in line with findings reported from Ethiopia east Africa [27], Ethiopia [23], India [32], SSA [11] and low and middle income [34]. The possible reason could be that low household wealth is connected to food insecurity, which prevents individuals from being well-nourished and increases their risk of micro nutrient deficiency including iron deficiency anemia. Additionally, the widespread occurrence of anemia may be due to the fact that households with lower wealth are less likely to have sufficient purchases of nutrient-rich food and access to healthcare services for women when they are ill.

This study showed that women who were unemployed are more likely to develop anemia than those who are employed. Similar finding were reported from a multilevel analysis in east Africa [27], Mauritania, Nigeria [35] and primary study in Senegal [12]. This may be explained by the fact that the woman's financial independence gives her more control over the decisions she makes regarding her follow-up care. An excellent degree of socioeconomic well-being acts as a buffer against anemia. Thus, anemia is a health issue associated with the nations' socioeconomic status. A higher socioeconomic status would improve nutritional status and act as a preventive factor against anemia.

This study also revealed being underweight was significantly associated with increased the likelihood of anemia among reproductive age women. This is in line with studies conducted in Indonesia [36], Ethiopia [23], SSA [11], and [13] Rwanda. Women who are underweight are more likely to develop anemia due to several factors. One reason is that they may not be consuming enough iron-rich foods in their diets. Another reason is that they may have lower total body iron stores due to lower muscle mass, which can result in lower ferritin levels [37]. Ferritin is a protein that stores iron in the body and helps to regulate its release. Having lower ferritin levels means that less iron is available to produce hemoglobin and maintain healthy red blood cells [38,39]. Therefore, it is crucial for underweight women to maintain a healthy weight and consume a balanced diet rich in iron and other essential nutrients to prevent anemia and promote overall health.

The current study reveal that the prevalence of anemia were reduced by 28% among women current users of hormonal contraceptives; these findings are consistent with those of other studies in Tanzania [39], Rwanda [13], twenty four SSA countries [40] and population based study in SSA [41]. This could be explained by hormonal contraception methods like the pill, patch, ring, and injection use synthetic hormones (estrogen and progestin) to prevent pregnancy. These hormones help regulate menstrual cycles, decrease the amount of blood lost during periods, and prevent ovulation altogether. As a result, women who use hormonal contraception methods are less likely to experience heavy menstrual bleeding and prolonged periods, which can lead to iron deficiency anemia [42,43].

The prevalence of be anemic among pregnant reproductive age women were 1.21 times higher compared to non-pregnant women. This is consistent with the findings reported in previous studies Senegal [12], Gambia, Indonesia [36], Democratic Republic of Congo [24], and Ethiopia [23]. During pregnancy there are factors that can increase the chances of women developing anemia. The changes, in the body during pregnancy such as an increase in blood volume can dilute blood cells and hemoglobin levels leading to a decrease in their levels and eventually causing anemia [44]. Additionally, pregnant women need nutrients, iron to produce extra red blood cells. Insufficient intake of nutrients, through diet having pregnancies or pregnancies that are closely spaced apart and certain complications can further strain the body's resources and raise the risk of anemia. It is crucial to screen for, monitor and manage anemia during pregnancy to ensure the well-being of both the mother and the fetus.

In this study, alcohol consumption decreases the likelihood of developing anemia among reproductive age women; in line with the study in Ethiopia [45] and Tanzanian adults[46]. This happens because of a few factors. One is that when we consume alcohol it has been found to increase the absorption of iron, from the food we eat. Additionally research has shown that alcohol can stimulate the production of erythropoietin, a hormone that tells our bone marrow to make blood cells [47]. Furthermore alcohol can also improve blood flow and circulation in our bodies, which helps deliver oxygen and nutrients to our tissues and organs [48]. It is important to note, however, that excessive alcohol consumption can have a negative impact on health and may increase the risk of developing other conditions such as liver disease, heart disease, and certain types of cancer. Therefore, moderation is key when it comes to alcohol consumption [49].

Reproductive-age women from eastern and Zanzibar administrative zones were significantly associated with higher odds of having anemia, whereas women from southern administrative zones had a lower prevalence of anemia. Several factors contribute to the variation in anemia prevalence rates across different administrative zones in Tanzania. These factors include socioeconomic factors such as income levels, poverty rates, and access to clean water, sanitation facilities, and healthcare services. Food security and dietary patterns also play a role, with variations in agricultural productivity and the availability of nutritious foods affecting anemia prevalence rates [50]. It is important to consider that these factors interact with each other, and multiple factors may contribute to the geographic variation of anemia in Tanzania.

Women living in communities with a high level of community poverty were more likely to suffer anemia than women who lived in communities with low levels of community poverty. This is consistent with another study SSA [11] and India [51]. The likelihood of anemia is heightened in areas with high community poverty due to a lack of access to nutritious food, inadequate sanitation and hygiene practices, limited healthcare, and essential services, as well as poor living conditions [52,53]. It is crucial to address community poverty and underlying socioeconomic factors in order to decrease anemia prevalence rates.

The high prevalence of anemia among reproductive-age women in Tanzania has significant implications for the health and development of the country. Addressing the root causes of

anemia, such as improving access to nutritious food, implementing effective interventions, and increasing awareness of anemia prevention, is crucial for the overall health and well-being of women and children in Tanzania.

## Strengths and limitations of the study

The present study is one of the few to report on the prevalence and predictors of anemia among reproductive women in Tanzania at the national level, making use of a sufficiently-sized sample that enhances the reliability of the data. The measurements were made using standard national tools and methods, and the data collection process was carried out by experts. However, because the data are cross-sectional, it is not possible to demonstrate the cause-and-effect link between the explanatory and outcome variables. Moreover, recall bias may exist because study participants were asked to recollect events that occurred five years or more prior to the survey.

## Conclusion

In conclusion, the prevalence of anemia among reproductive-age women in Tanzania is significantly higher compared to the global prevalence reported by the WHO. This finding underscores the urgent need for concerted efforts to address this public health issue in Tanzania. The high prevalence of anemia among reproductive-age women not only poses a significant threat to their health but also has severe implications for maternal and child health outcomes. Interventions targeting the improvement of nutrition, access to healthcare services, and education on anemia prevention and management should be prioritized to effectively reduce the burden of anemia among reproductive-age women in Tanzania. Additionally, further research is warranted to explore the unique factors contributing to this high prevalence and tailor interventions accordingly. By implementing these measures, we can strive to improve the health and well-being of women in Tanzania, ensuring better maternal and child health outcomes.

## Acknowledgments

We would like to thank the measure DHS program for providing the datasets.

## Author Contributions

**Conceptualization:** Gizachew Ambaw Kassie, Aklilu Habte Hailegebireal, Amanuel Yosef Gebrekidan, Beshada Zerfu Woldegeorgis, Getachew Asmare Adella, Yordanos Sisay Asgedom.

**Data curation:** Gizachew Ambaw Kassie, Aklilu Habte Hailegebireal, Amanuel Yosef Gebrekidan, Beshada Zerfu Woldegeorgis, Getachew Asmare Adella, Kirubel Eshetu Haile, Yordanos Sisay Asgedom.

**Formal analysis:** Gizachew Ambaw Kassie, Amanuel Yosef Gebrekidan, Kirubel Eshetu Haile, Yordanos Sisay Asgedom.

**Funding acquisition:** Gizachew Ambaw Kassie, Beshada Zerfu Woldegeorgis, Getachew Asmare Adella.

**Investigation:** Gizachew Ambaw Kassie, Amanuel Yosef Gebrekidan, Kirubel Eshetu Haile, Yordanos Sisay Asgedom.

**Methodology:** Gizachew Ambaw Kassie, Aklilu Habte Hailegebireal, Beshada Zerfu Woldegeorgis, Getachew Asmare Adella, Kirubel Eshetu Haile, Yordanos Sisay Asgedom.

**Project administration:** Gizachew Ambaw Kassie, Beshada Zerfu Woldegeorgis.

**Resources:** Gizachew Ambaw Kassie, Amanuel Yosef Gebrekidan, Getachew Asmare Adella, Kirubel Eshetu Haile, Yordanos Sisay Asgedom.

**Software:** Gizachew Ambaw Kassie, Beshada Zerfu Woldegeorgis, Yordanos Sisay Asgedom.

**Supervision:** Aklilu Habte Hailegebireal.

**Validation:** Aklilu Habte Hailegebireal, Amanuel Yosef Gebrekidan, Beshada Zerfu Woldegeorgis, Kirubel Eshetu Haile, Yordanos Sisay Asgedom.

**Visualization:** Gizachew Ambaw Kassie, Amanuel Yosef Gebrekidan, Getachew Asmare Adella, Kirubel Eshetu Haile, Yordanos Sisay Asgedom.

**Writing – original draft:** Gizachew Ambaw Kassie, Beshada Zerfu Woldegeorgis, Getachew Asmare Adella, Yordanos Sisay Asgedom.

**Writing – review & editing:** Gizachew Ambaw Kassie, Aklilu Habte Hailegebireal, Amanuel Yosef Gebrekidan, Beshada Zerfu Woldegeorgis, Getachew Asmare Adella, Kirubel Eshetu Haile, Yordanos Sisay Asgedom.

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
