## [Decision Letter · Decision Letter 0]

30 Apr 2024

PONE-D-24-03494Anemia Status and its Determinants among Reproductive-age Women in Tanzania: A Multi-level analysis of Tanzanian Demographic and Health Survey DataPLOS ONE

Dear Dr. Kassie,

Thank you for submitting your manuscript to PLOS ONE. After careful consideration, we feel that it has merit but does not fully meet PLOS ONE’s publication criteria as it currently stands. Therefore, we invite you to submit a revised version of the manuscript that addresses the points raised during the review process.

I would like to sincerely apologise for the delay you have incurred with your submission. It has been exceptionally difficult to secure reviewers to evaluate your study. We have now received two completed reviews; the comments are available below. The reviewers have raised significant scientific concerns about the study that need to be addressed in a revision.

Please revise the manuscript to address all the reviewer's comments in a point-by-point response in order to ensure it is meeting the journal's publication criteria. Please note that the revised manuscript will need to undergo further review, we thus cannot at this point anticipate the outcome of the evaluation process.

We look forward to receiving your revised manuscript.

Kind regards,

Miquel Vall-llosera Camps

Staff Editor

PLOS ONE

Journal Requirements:

https://doi.org/10.1371/journal.pone.0296451

In your revision ensure you cite all your sources (including your own works), and quote or rephrase any duplicated text outside the methods section. Further consideration is dependent on these concerns being addressed.

Reviewers' comments:

Reviewer's Responses to Questions

**Comments to the Author**

1. Is the manuscript technically sound, and do the data support the conclusions?

Reviewer #1: Partly

Reviewer #2: Yes

2. Has the statistical analysis been performed appropriately and rigorously? 

Reviewer #1: Yes

Reviewer #2: Yes

3. Have the authors made all data underlying the findings in their manuscript fully available?

Reviewer #1: Yes

Reviewer #2: Yes

4. Is the manuscript presented in an intelligible fashion and written in standard English?

Reviewer #1: No

Reviewer #2: Yes

5. Review Comments to the Author

Reviewer #1: This study aimed to assess the burden of anemia and its determinants among reproductive aged women using the Tanzanian Demographic and Health Survey data. I made few comments that can improve the readability and quality of the manuscript mentioned below:

1) The background section needs to be improved, specifically the rationale of the study, though author tried to make the rationale clear.

2) Line 111-120: You need citation for classification of anemia. Just mentioning anemia level classification from the previous classification is not sufficient. Why you don’t follow the WHO criteria? Additionally, you need to explain what types of anemia you measured (like iron deficiency anemia)? See https://www.cochranelibrary.com/cdsr/doi/10.1002/14651858.CD013092.pub2/full.

3) Line 121-133: You need further explanation and citation for several independent variables how you categorized them (e.g., BMI and so on).

4) Line 143: You mentioned that “logistic regression model was employed in our study since the anemia prevalence was high….”. How did you define high prevalence requires further explanation.

5) Could you please add a sampling framework of the study?

6) What did you mean no education? Explain in your method section.

7) Line 214-218 seems to be redundant as you explained it in the method section. Drop it.

8) Line 222: What is the meaning of significant association with anemia?

9) Line 224-240: You mentioned that “…. Higher odds of a higher level of anemia….” which is completely inappropriate explanation. Also, you reported multivariate results in percentage and in times, please follow one style.

10) In discussion section, it is suggested to avoid quantitative results with Cis that you reported in result section. Please revise it.

11) Please provide a clear script of STATA code (do file) as an appendix.

Reviewer #2: This is an interesting paper and utilized most recent data of Tanzanian DHS to assess anemia status and its determinants among reproductive-age women.

Methodology

1. In the study variables and measurements section you showed how the dependent variable was measured but it was not cited.

2. The prevalence of anemia among women of reproductive age in Tanzania was reported 42.02%. Considering the cross-sectional nature of the data and the large prevalence of the outcome variable, odds ratio might overestimate the association between the dependent and independent variables. Therefore in my view multilevel Poisson regression analysis with robust variance would be preferable.

6. PLOS authors have the option to publish the peer review history of their article (what does this mean?). If published, this will include your full peer review and any attached files.

Reviewer #1: **Yes: **Md. Obaidur Rahman

Reviewer #2: No

---

## [Author Response · Author response to Decision Letter 0]

30 May 2024

Point by point response to reviewers

Editorial comment: Please revise the manuscript to address all the reviewer's comments in a point-by-point response in order to ensure it is meeting the journal's publication criteria. Please note that the revised manuscript will need to undergo further review, we thus cannot at this point anticipate the outcome of the evaluation process. Please include the following items when submitting your revised manuscript: A rebuttal letter that responds to each point raised by the academic editor and reviewer(s). You should upload this letter as a separate file labeled 'Response to Reviewers'. A marked-up copy of your manuscript that highlights changes made to the original version. You should upload this as a separate file labeled 'Revised Manuscript with Track Changes'. An unmarked version of your revised paper without tracked changes. You should upload this as a separate file labeled 'Manuscript'.

Response: Dear academic editor, we would like to thank you for your time and effort in facilitating our manuscript revision. In the Response to Reviewers, we copy each comment and provide the response hereunder. We also provide a marked-up copy of the manuscript that indicates tracked changes made to the original version, and this is uploaded as a separate file labeled “Revised Manuscript with Track Changes” and an unmarked version of our revised paper without tracked changes uploaded as a separate file labeled 'Manuscript'.

Editorial comment 2: Journal Requirements: When submitting your revision, we need you to address these additional requirements. Please ensure that your manuscript meets PLOS ONE's style requirements, including those for file naming. We noticed you have some minor occurrence of overlapping text with the following previous publication(s), which needs to be addressed: https://doi.org/10.1371/journal.pone.0296451 In your revision ensure you cite all your sources (including your own works), and quote or rephrase any duplicated text outside the methods section. Further consideration is dependent on these concerns being addressed.

Response: Thank you for your comments and feedback, we have carefully reviewed and addressed all of your concerns. We have made sure that our manuscript meets PLOS ONE's style requirements, including file naming. Additionally, we have carefully checked the manuscript for any overlapping text with the previous publication(s) you mentioned and have addressed them. Page #8, line #1-38

Reviewers' comments:

Review Comments to the Author

Reviewer #1: 

General comment: This study aimed to assess the burden of anemia and its determinants among reproductive aged women using the Tanzanian Demographic and Health Survey data. I made few comments that can improve the readability and quality of the manuscript mentioned below:

Response: Dear reviewer, we would like to express our gratitude for the time you have taken to insightful and constructive comments on our manuscript, and we appreciate the effort you have put in to review our paper. We have carefully considered all your comments and suggestions. Page #2-28

Comment 1: The background section needs to be improved, specifically the rationale of the study, though author tried to make the rationale clear.

Response: Thank you for your feedback. We acknowledge that the background section, particularly the rationale of the study, needs improvement. We have revised and provided a clearer and more compelling context for our research. Page #3-5

Comment 2: Line 111-120: You need citation for classification of anemia. Just mentioning anemia level classification from the previous classification is not sufficient. Why you don’t follow the WHO criteria? Additionally, you need to explain what types of anemia you measured (like iron deficiency anemia)? Seehttps://www.cochranelibrary.com/cdsr/doi/10.1002/14651858.CD013092.pub2/full.

Response: Thank you for your valuable comment. We now provide a citation for the classification of anemia. Specifically, we have incorporated the World Health Organization (WHO) criteria to enhance the clarity of our anemia classification. Regarding types of Anemia we have included an explanation of the types of anemia measured in the revised version of our manuscript. Page #7, line #149-154

Comment 3: Line 121-133: You need further explanation and citation for several independent variables how you categorized them (e.g., BMI and so on).

Response: Thank you for your recommendation. We have further explained our variable measurements and provided citations for each operation definition in the revised manuscript. Page #8-9, line #172-188 

Comment 4: Line 143: You mentioned that “logistic regression model was employed in our study since the anemia prevalence was high….”. How did you define high prevalence requires further explanation. 

Response: Thank you for your feedback. We apologize for the editorial error. We have rephrased it in the revised version of our manuscripts. Page #9, line #198

Comment 5: Could you please add a sampling framework of the study?

Response: Thank you for your comment. We have taken your feedback into consideration and have now included a more detailed description of the sampling framework in the methodology section of the paper. Additionally, we have created a diagrammatic representation of the sampling process to provide a clearer understanding of how we selected our study participants. We hope that these additions will enhance the comprehensibility of our research and will address your concerns regarding the sampling framework. Thank you for your valuable feedback and suggestions for improving our work. Page #6-7, line #125-145

Comment 6: What did you mean no education? Explain in your method section.

Response: In our study, when we mentioned “no education,” we were referring to individuals who did not complete any formal education. We apologize for any confusion and appreciate the opportunity to clarify this point. In the revised method section, we have elaborated on this definition to ensure transparency and accuracy.

Comment 7: Line 214-218 seems to be redundant as you explained it in the method section. Drop it.

Response: Thank you for your insightful comment. We appreciate your attention to detail. Based on your suggestion, we have removed it from the manuscript as they were redundant with the information already provided in the method section. Page #15, line #267-271

Comment 8: Line 222: What is the meaning of significant association with anemia?

Response: Thank you for your question for clarification. The term “significant association” in our study refers to a statistically meaningful relationship between two variables. Specifically, when we say that there is a significant association with anemia, we mean that the presence of anemia is related to another factor (such as a risk factor, exposure, or outcome) in a way that is unlikely to occur by chance alone. Page #15, line #275

Comment 9: Line 224-240: You mentioned that “…. Higher odds of a higher level of anemia….” which is completely inappropriate explanation. Also, you reported multivariate results in percentage and in times, please follow one style. 

Response: Dear reviewer, we apologize for any confusion. In the revised manuscript, we have revised the interpretation to avoid ambiguity. Additionally, we have ensured consistency in reporting multivariate results by using a single style (either percentages or times). However, it is important to note that the interpretation of the odds ratio depends on whether the value is greater than 1 or less than 1. Therefore, it is crucial to consider the direction of the odds ratio when interpreting the results of a study. Thank you for bringing this important point to our attention. Page #15-16, line #277-296

Comment 10: In discussion section, it is suggested to avoid quantitative results with Cis that you reported in result section. Please revise it.

Response: Dear reviewer, thank you for you insightful comments. We have removed the confidence interval based on your suggestion. We have been mentioned it in the discussion section for comparison purpose our result with other similar findings. Page #18, line #308-310

Comment 11: Please provide a clear script of STATA code (do file) as an appendix.

Response: We appreciate the reviewer’s feedback. In response, we have uploaded comprehensive general STATA commands and the data set as “other” in the submission system manuscript. 

Reviewer #2:

General comment: This is an interesting paper and utilized most recent data of Tanzanian DHS to assess anemia status and its determinants among reproductive-age women.

Response: Response: Dear reviewer, thank you for devoting your time to review our manuscript and we appreciate your deep insight of the manuscript. We have carefully considered all your comments and suggestions.

Methodology

Comment 1: In the study variables and measurements section you showed how the dependent variable was measured but it was not cited.

Response: Thank you for your feedback. We apologize for the oversight. In the revised manuscript, we have appropriately cited the references related to the measurement of the dependent variable in the study variables and measurements section. Page #8-9, line #172-188

Comment 2: The prevalence of anemia among women of reproductive age in Tanzania was reported 42.02%. Considering the cross-sectional nature of the data and the large prevalence of the outcome variable, odds ratio might overestimate the association between the dependent and independent variables. Therefore in my view multilevel Poisson regression analysis with robust variance would be preferable.

Response: Thank you for your valuable feedback regarding the statistical method used in our study. We appreciate your concern about the potential for odds ratios to overestimate the association between the dependent and independent variables. However, we carefully considered the characteristics of our dataset, research question, and study design and determined that odds ratios were the most appropriate statistical method to achieve our research objectives. Our study has a large sample size, which mitigates the influence of high prevalence of anemia among Tanzanian women of reproductive age while ensuring the precision of our odds ratio estimates. Moreover, our focus was on exploring relationships between anemia and various independent factors and not to forecast the prevalence rates of anemia. In this context, odds ratios are a robust measure that captures the strength and direction of these associations. Nonetheless, we appreciate your suggestion to use multilevel Poisson regression analysis with robust variance in future studies and will consider it for future research endeavors.

---

## [Decision Letter · Decision Letter 1]

26 Jul 2024

PONE-D-24-03494R1Anemia Status and its Determinants among Reproductive-age Women in Tanzania: A Multi-level analysis of Tanzanian Demographic and Health Survey DataPLOS ONE

Dear Dr. Kassie,

Thank you for submitting your manuscript to PLOS ONE. After careful consideration, we feel that it has merit but does not fully meet PLOS ONE’s publication criteria as it currently stands. Therefore, we invite you to submit a revised version of the manuscript that addresses the points raised during the review process.

The manuscript was improved; however, significant scientific concerns remain. Please revise the manuscript to address all the reviewer's comments in a point-by-point response. In addition, a revision of the statistical analysis should be performed as recommended by the reviewer in the first revision round.  The authors’ reasons for not run a more appropriate statistical approach are not supported by the literature – the multilevel Poisson regression models are more appropriate for cross-sectional study design.

We look forward to receiving your revised manuscript.

Kind regards,

Marly A. Cardoso, Ph.D.

Academic Editor

PLOS ONE

Additional Editor Comments:

The authors should revise the statistical analysis as recommended before by the reviewer as follows "The prevalence of anemia among women of reproductive age in Tanzania was reported 42.02%. Considering the cross-sectional nature of the data and the large prevalence of the outcome variable, odds ratio might overestimate the association between the dependent and independent variables. Therefore in my view multilevel Poisson regression analysis with robust variance would be preferable".

Reviewers' comments:

Reviewer's Responses to Questions

**Comments to the Author**

1. If the authors have adequately addressed your comments raised in a previous round of review and you feel that this manuscript is now acceptable for publication, you may indicate that here to bypass the “Comments to the Author” section, enter your conflict of interest statement in the “Confidential to Editor” section, and submit your "Accept" recommendation.

Reviewer #3: (No Response)

2. Is the manuscript technically sound, and do the data support the conclusions?

Reviewer #3: Yes

3. Has the statistical analysis been performed appropriately and rigorously? 

Reviewer #3: Yes

4. Have the authors made all data underlying the findings in their manuscript fully available?

Reviewer #3: Yes

5. Is the manuscript presented in an intelligible fashion and written in standard English?

Reviewer #3: Yes

6. Review Comments to the Author

Reviewer #3: Thank you for the opportunity to read this interesting manuscript that assessed the prevalence of anemia and its determinants among reproductive aged women using data from the Tanzanian Demographic and Health Survey. In my opinion, the manuscript is a good piece of work, well structured and easy to follow and understand. The introduction section is well presented, provides a contextualization of the topic addressed and presents the justification for the study. The methods section is well detailed and can be replicated. However, I did not identify the inclusion and exclusion criteria for the study. Another observation is about the cutoff points adopted to assess the severity of anemia. According to the WHO document (ref 18), the cutoff points described in lines 140-142 are for pregnant women. Furthermore, the information present in lines 140-142 is different from the information present in lines 169-170. I suggest checking it out.

In my opinion, the statistical analyses are sound and the results are robust. In the results section, I suggest presenting the results in the order they appear in the tables. Also, check the administrative zones associated with anemia (line 253). The central administrative zone is not associated with anemia according to Table 4.

In the discussion section the authors explain the results and make relevant comparisons with previous studies, and the conclusions are well-stated.

I have minor suggestions for the authors to improve the quality of this manuscript:

- Line 64-65: I suggest including in the text the year in which this Tanzania Demographic and Health Survey was carried out.

- Line 77: I suggest reviewing the text. Is there a period after “status”?

- Line 80: Include a period at the end of the sentence.

- Line 140: I suggest presenting the acronym (Hgb) in full first.

- Line 141: I suggest correcting to “less than 7 g/dL” for severe anemia.

- Line 164: I think the topic “Operational definition”, when describing anemia, is a bit redundant. Additionally, I suggest that the BMI classification be in the “independent variables” topic.

- Lines 211 and 224: The values presented in % are different from the table. I suggest checking.

- Table 4 (Model III): I suggest standardizing the number of “middle household wealth status”.

- Line 255-256: check the data on the “wealth index of the poorest families”. It is not in accordance with table 4.

- Line 261: Please check the 95%CI of pregnancy. It is not in accordance with table 4.

- Line 261: Please correct: “body mass index of <18.5 kg/m2”

- Table 4: Why are the Southern Administrative Zone values in bold?

7. PLOS authors have the option to publish the peer review history of their article (what does this mean?). If published, this will include your full peer review and any attached files.

Reviewer #3: No

---

## [Author Response · Author response to Decision Letter 1]

10 Sep 2024

Point by point response to reviewers

Editorial comment: Thank you for submitting your manuscript to PLOS ONE. After careful consideration, we feel that it has merit but does not fully meet PLOS ONE’s publication criteria as it currently stands. Therefore, we invite you to submit a revised version of the manuscript that addresses the points raised during the review process. The manuscript was improved; however, significant scientific concerns remain. Please revise the manuscript to address all the reviewer's comments in a point-by-point response. In addition, a revision of the statistical analysis should be performed as recommended by the reviewer in the first revision round. The authors’ reasons for not run a more appropriate statistical approach are not supported by the literature – the multilevel Poisson regression models are more appropriate for cross-sectional study design. 

Response: Dear academic editor, we would like to thank you for your time and effort in facilitating our manuscript revision. Thank you for your feedback on our manuscript. We have tried to address all your and the reviewers’ comments in revised the manuscript. Specifically, we have clarified our methodology and revised the statistical analysis using multilevel Poisson regression models, as recommended. We believe these changes have significantly improved our manuscript. Thank you for considering our revised submission. Page #17-20

Editorial comment 2: The authors should revise the statistical analysis as recommended before by the reviewer as follows "The prevalence of anemia among women of reproductive age in Tanzania was reported 42.02%. Considering the cross-sectional nature of the data and the large prevalence of the outcome variable, odds ratio might overestimate the association between the dependent and independent variables. Therefore in my view multilevel Poisson regression analysis with robust variance would be preferable"

Response: Thank you for your comments and feedback, we have carefully reviewed and addressed all of your concerns. We have revised the statistical analysis as recommended by the reviewer. We have employed multilevel Poisson regression analysis with robust variance to better account for the cross-sectional nature of the data and the high prevalence of the outcome variable. Page #2-25

Review Comments to the Author

Reviewer #3: 

General comment: Thank you for the opportunity to read this interesting manuscript that assessed the prevalence of anemia and its determinants among reproductive aged women using data from the Tanzanian Demographic and Health Survey. In my opinion, the manuscript is a good piece of work, well-structured and easy to follow and understand. The introduction section is well presented, provides a contextualization of the topic addressed and presents the justification for the study. The methods section is well detailed and can be replicated. However, I did not identify the inclusion and exclusion criteria for the study. Another observation is about the cutoff points adopted to assess the severity of anemia. According to the WHO document (ref 18), the cutoff points described in lines 140-142 are for pregnant women. Furthermore, the information present in lines 140-142 is different from the information present in lines 169-170. I suggest checking it out. In my opinion, the statistical analyses are sound and the results are robust. In the results section, I suggest presenting the results in the order they appear in the tables. Also, check the administrative zones associated with anemia (line 253). The central administrative zone is not associated with anemia according to Table 4. In the discussion section the authors explain the results and make relevant comparisons with previous studies, and the conclusions are well-stated.

Response: Thank you for the time and effort you have invested in reviewing our manuscript. We highly appreciate your valuable feedback and constructive criticisms on the manuscript. We have now revised the paper based on your comments and suggestions. Page #2-24

Comment 1: Line 64-65: I suggest including in the text the year in which this Tanzania Demographic and Health Survey was carried out.

Response: Thank you for your valuable feedback. We have included the year in which the Tanzania Demographic and Health Survey was conducted in the revised version of our manuscript. Specifically, we have noted that the survey was carried out in 2015/16. Page #3, line #65

Comment 2: Line 77: I suggest reviewing the text. Is there a period after “status”?

Response: Thank you for your suggestion. We have reviewed the text and addressed the issue regarding the period after “status.” The necessary correction has been made in the revised version of our manuscript. Page #4, line #78

Comment 3: Line 80: Include a period at the end of the sentence.

Response: Thank you for your comment and we have made correction accordingly on the revised version of the manuscript. Page #4, line #81

Comment 4: Line 140: I suggest presenting the acronym (Hgb) in full first.

Response: Thank you for pointing out it I have now presented the acronym “Hgb” in full as “hemoglobin” at its first occurrence in the document. This should enhance clarity for all readers. Page #6, line #136

Comment 5: Line 141: I suggest correcting to “less than 7 g/dL” for severe anemia.

Response: Thank you for your feedback. I have accepted your comment and made the necessary corrections accordingly. Page #7, line #141-148

Comment 6: - Line 164: I think the topic “Operational definition”, when describing anemia, is a bit redundant. Additionally, I suggest that the BMI classification be in the “independent variables” topic.

Response: We appreciate your attention to detail. Based on your suggestion, we have removed it optional definition and merge with the independent variable section accordingly. Page #18, line #169-182

Comment 7:-Lines 211 and 224: The values presented in % are different from the table. I suggest checking.

Response: Thank you for pointing this out. It was an editorial error we corrected it on the revised manuscript. Page #11, line #234

Comment 8- Table 4 (Model III): I suggest standardizing the number of “middle household wealth status”.

Response: Thank you very much for your valuable feedback and suggestion on our manuscript. All of our important findings have been presented in line with those reported in the TDHS results. Page #18

Comment 9: - Line 255-256: check the data on the “wealth index of the poorest families”. It is not in accordance with table 4.

Response: Dear reviewer, thank you for pointing out it. We have corrected in the revised manuscripts. Page #18

Comment 10: - Line 261: Please check the 95%CI of pregnancy. It is not in accordance with table 4.

Response: Thank you for your comment; we apologize for the incontinence we have corrected in the revised version. Page #16, line #286

Comment 11:- Line 261: Please correct: “body mass index of <18.5 kg/m2”

Response: Dear reviewer, thank you for bringing that error to our attention. We apologize for the oversight. It was an editorial error made during paraphrasing and write-up of our results. We have made an extensive revision of our manuscript and corrected all editorial and typo errors. Page #16, line #279

Comment 12: - Table 4: Why are the Southern Administrative Zone values in bold?

Response: Thank you again for bringing these errors. We accepted the comments and we have made the necessary correction in the revised version of our manuscript. Page #19

---

## [Editor Report · Decision Letter 2]

13 Sep 2024

Anemia Status and its Determinants among Reproductive-age Women in Tanzania: A Multi-level analysis of Tanzanian Demographic and Health Survey Data

PONE-D-24-03494R2

Dear Dr. Kassie,

We’re pleased to inform you that your manuscript has been judged scientifically suitable for publication and will be formally accepted for publication once it meets all outstanding technical requirements.

Kind regards,

Marly A. Cardoso, Ph.D.

Academic Editor

PLOS ONE
---

## [Editor Report · Acceptance letter]

19 Sep 2024

PONE-D-24-03494R2 

PLOS ONE

Dear Dr. Kassie, 

I'm pleased to inform you that your manuscript has been deemed suitable for publication in PLOS ONE. Congratulations! Your manuscript is now being handed over to our production team.

Kind regards, 

on behalf of

Dr. Marly A. Cardoso 

Academic Editor

PLOS ONE